# Metabolic Pathways in Breast Cancer Reprograming: An Insight to Non-Coding RNAs

**DOI:** 10.3390/cells11192973

**Published:** 2022-09-23

**Authors:** Fereydoon Abedi-Gaballu, Elham Kamal Kazemi, Seyed Ahmad Salehzadeh, Behnaz Mansoori, Farhad Eslami, Ali Emami, Gholamreza Dehghan, Behzad Baradaran, Behzad Mansoori, William C. Cho

**Affiliations:** 1Immunology Research Center, Tabriz University of Medical Sciences, Tabriz 51666-14731, Iran; 2Department of Biology, Faculty of Natural Sciences, University of Tabriz, Tabriz 51666-16471, Iran; 3Department of Medicinal Biotechnology, Faculty of Medical Sciences, Tarbiat Modares University, Tehran 175-14115, Iran; 4Department of Molecular Genetics, Faculty of Biological Sciences, Tarbiat Modares University, Tehran 175-14115, Iran; 5Cellular and Molecular Oncogenesis Program, The Wistar Institute, Philadelphia, PA 19104, USA; 6Department of Clinical Oncology, Queen Elizabeth Hospital, Hong Kong, China

**Keywords:** breast cancer, reprograming, non-coding RNA, metabolic pathways

## Abstract

Cancer cells reprogram their metabolisms to achieve high energetic requirements and produce precursors that facilitate uncontrolled cell proliferation. Metabolic reprograming involves not only the dysregulation in glucose-metabolizing regulatory enzymes, but also the enzymes engaging in the lipid and amino acid metabolisms. Nevertheless, the underlying regulatory mechanisms of reprograming are not fully understood. Non-coding RNAs (ncRNAs) as functional RNA molecules cannot translate into proteins, but they do play a regulatory role in gene expression. Moreover, ncRNAs have been demonstrated to be implicated in the metabolic modulations in breast cancer (BC) by regulating the metabolic-related enzymes. Here, we will focus on the regulatory involvement of *ncRNAs (microRNA, circular RNA and long ncRNA)* in BC metabolism, including glucose, lipid and glutamine metabolism. Investigation of this aspect may not only alter the approaches of BC diagnosis and prognosis, but may also open a new avenue in using ncRNA-based therapeutics for BC treatment by targeting different metabolic pathways.

## 1. Introduction

Cancer metabolism, based on energy reprograming, is distinguished and is confirmed as one of cancer’s hallmarks [1,2]. Cancer cells are able to keep their metabolisms high to reach the energy requirements associated with sustaining uncontrolled cell proliferation [3]. Therefore, great changes in cancer cell metabolism not only involves generating anabolic macromolecules, which are implicated in the development of cancers, including breast tumors, but also the need to maintain a transformed state and promote cell proliferation [4]. Some of the dysregulation in the metabolism of breast cancer (BC) biology includes (a) a glycolysis process that performs significantly in BC due to a notable overexertion of hexokinase 2 (HK2) which provides the requirements of cancer cells even in the presence of high levels of oxygen (Warburg effect) [5]. (b) Glutaminolysis which widely exists in some tumors and creates downstream intermediates such as glutamate and α-ketoglutarate which will be used to synthesize cholesterol, amino acids, lipids and other underlying metabolites [6]. (c) The generation of nicotinamide adenine dinucleotide phosphate (NADPH) and ribonucleotide intermediates through the pentose phosphate pathway (PPP) in cancer cells could facilitate the glycolytic pathway, lipid biosynthesis, neutralize the reactive oxidative stress and provide ribose-5-phosphate to produce nucleotide. (d) The raising of lipid metabolism, especially *de novo* lipogenesis in BC cells, which is needed to generate new organelles or cells [7]. (e) Fatty acid oxidations and mitochondrial biogenesis are other pathways that actively occur to enhance BC cells’ survival and proliferation and provide the necessary energy and precursors for tumors [8,9]. Therefore, despite the apparent role of the metabolic changes (mentioned above) in developing BC, identifying numerous gene expressions and proteins of human breast tumors, there is still relatively little known about the regulation of the metabolism changes that participate in BC development. On the other hand, metabolic reprograming is dependent on the alteration of several pathways which are regulated by tumor suppressor genes and oncogenes, including phosphatase and tensin homolog (PTEN), p53, avian myelocytomatosis viral oncogene homolog (MYC) and hypoxia-inducible factor (HIF) [10]. Moreover, it emerges that the activation and inactivation of oncogenes and tumor suppressor genes could be regulated by non-coding RNAs (ncRNAs) such as small ncRNA and long ncRNA (lncRNA) [11]. For example, small regulatory ncRNAs, such as miRNAs and lncRNA, are able to reduce protein–coding gene expression via targeting their related *mRNA* which eventually inhibits the translation process [12]. Moreover, malfunction in miRNA expression has been involved in the upregulating or downregulating of oncogenes or tumor suppressors in BC, which impress cancer progression, initiation, metastasis etc. [13]. In this article, we discuss the BC metabolism changes pathways that are affected by the activation and inactivation of oncogenes and tumor suppressor ncRNAs. In addition, we describe the altered cancer cell metabolism and the use of ncRNAs as the modulators of specific metabolic pathways in BC. Finally, we cover the promising capabilities of ncRNA prognosis and therapeutic agents in BC.

## 2. Non-Coding RNAs

*ncRNAs* are classified based on their size as two major types, including long non-coding RNA (*lncRNAs)* and small *ncRNAs.* They never translate to the protein, while they are playing a key role in regulating of expression of target genes. The *lncRNAs* are a type of transcription factor with a length >200 bp that cannot end in proteins, but that modulate the gene’s expression at different levels in the form of RNA [14] and which were introduced as a regulator of crucial metabolism compartments involving enzyme expression. They have regulatory roles in the metabolizing or anabolizing of the glucose [15], lipids [16] and amino acids [17] that are implicated in cell growth, cell differentiation and cell death hemostasis, such as apoptosis. Unlike *lncRNAs*, the length of *small ncRNAs* is less than 200 nts, which includes *microRNAs (miRNAs), Piwi-interacting RNA, (piRNAs), small interfering RNAs (siRNAs)* and transcription initiation *RNAs (tRNAs*) [18]. Among *small ncRNAs, miRNAs* are 19–22 nts in length and are involved in regulating over 30% of pivotal genes, which are able to impress some of the biological processes including metabolic hemostasis, proliferation and growth [19]. *Circular RNAs (circRNAs)* are another member of *small ncRNAs* that act as regulatory *ncRNAs*, splicing regulators, protein scaffolds and *miRNA* sponges in a wide variety of human cancers [20]. C*ircRNAs* belong to the family of *lncRNAs* with sizes of 100–200 nts [21], and are implicated in almost all cellular functions, including cell proliferation, apoptosis-mediated cell death, autophagy and cell cycle arrest. Generally, it was demonstrated that some *ncRNAs* act as oncogenic or tumor suppressor genes (TSG) during carcinogenesis, which allow them to employ as diagnostic and prognostic indicators in BC patients after certain treatments [22,23].

## 3. Breast Cancer

BC is classified as one of the most common cancers in women and is responsible for more than 450,000 deaths each year around the world [24]. It is a heterogeneous disease, with six molecular subtypes, including luminal A, tumors enriched with human epidermal growth factor receptor 2 (HER-2), luminal B, claudin-low subtype, basal-like and normal-like [25]. In BC, reprogramed metabolic pathways develop to sustain macromolecular biosynthesis or energy production. It is well known that the metabolic properties of cancer cells are different than those from normal cells and are associated with performing more aerobic glycolysis, increased fatty acid synthesis and increased glutamine metabolism. Therefore, cancer cells are closely connected with metabolic disorders such as the alteration in the expression of key enzymes that governs the metabolic pathway. Malfunction in each metabolic enzyme could increase cancer development, which are widely affected by oncogenes and tumor suppressors.

## 4. *Non-Coding RNAs* as a Regulator of Metabolic Reprograming in BC

Cancer cells can use metabolic reprograming to produce macromolecules which are vital for their proliferation and survival [26]. Induction of the metabolic reprograming in cancer cells is carried out by a complex interaction of regulatory pathways including 5′ AMP-activated protein kinase (AMPK), mTOR, phosphatidylinositide 3-kinase (PI3K), protein kinase B (Akt) and PTEN. Many *ncRNAs* are involved in regulating lipids, glucose and the metabolisms of amino acids through the regulation of gene transcriptions and key enzymes that are necessary for metabolic pathways [27]. It is worth noting that there are 45,000 target sites for *ncRNAs* in the human genome which have a regulating role in up to 60% of human genes. *ncRNAs* are widely involved in the modulation of different biological events. Mechanistically, they could modulate cellular metabolism by directly affecting and regulating mRNA and the protein expression of metabolic key enzymes or indirectly by modulating the regulation of effectors such as transcription factors, which adjust the synthesis of metabolic enzymes [28].

## 5. *ncRNAs* Function as a Regulator of Glucose Metabolisms Reprograming in BC

It is more widely accepted that glucose has a key role in cancer development [29]. Under optimal concentration of oxygen, the normal cells metabolize glucose to make carbon dioxide via a tricarboxylic acid cycle (TCA) which provides 30 or 32 moles of adenosine triphosphate (ATP) per mole of glucose and a small level of lactate under oxidative phosphorylation [30]. It is worth commenting that in hypoxic conditions, the normal cells produce a higher amount of lactic acid through anaerobic glycolysis [31]. For the first time, Otto Warburg has suggested that tumor cells provide for their energy demands primarily from aerobic glycolysis when the oxygen level is limited. Surprisingly, it has been reported that glucose metabolic reprograming is the particulate observable evidence between aerobic glycolysis and oxidative phosphorylation in cancer cells [32]. However, cancer cells would use aerobic glycolysis with reducing mitochondrial oxidative phosphorylation to produce their energy demands even when the oxygen level is sufficient [33]. On the other hand, to compensate for the final reduction in ATP productions, cancer cells could adopt their mechanisms to increase their glucose uptake and utilizations. For this purpose, cancer cells highly express GLUT transporters (as main glucose uptake regulators) to increase glucose internalizations. Therefore, it seems that modulation of GLUT expressions because of their involvement in reducing glucose consumption may be a very helpful and effective strategy to control cancer cell proliferation, invasion and metastasis. Recently, *ncRNA* segments, including *lncRNA, miRNA* and *siRNA*, have been tested to regulate the GLUT expressions in metabolic reprogramed breast cancer [15,34,35]. Based on recent reports, *ncRNAs* often exert their regulatory roles via the direct modulation of key molecules (transporters, enzymes and kinases) or indirectly by regulating the expression of important transcription factors in reprogramed breast cancer.

### 5.1. ncRNAs Regulate GLUT Expression to Control Glucose Cellular Uptake and Metabolism

Na^+^-coupled glucose transporters (SGLT) and glucose transporters (GLUTs), as glucose carrier proteins in the membrane, are responsible for regulating glucose across the plasma cell membrane [34]. GLUTs are the most important proteins due to their regulatory role on glucose, catabolizing routes such as glycolysis by modulating glucose uptake in several tumors. As glucose is the main energy source for cancer cells, its cellular uptake is regularly modulated by some special GLUTs. On the other hand, cancer cells, by virtue of upregulating those proteins, tend to uptake glucose and provide their energy demands [35]. According to previous reports, among 14 members of GLUTs, GLUT1-4 and GLUT12 have a crucial regulatory role in maintaining the balance of glucose concentration on both sides of the plasma membrane in breast cancer [36,37]. Many studies reported that various types of *ncRNAs* can regulate glucose transport by adjusting the GLUTs expression.

miRNAs, as small *ncRNAs*, also play a key role in controlling glucose input by regulating the GLUTs. For example, analyzing the *miRNA–mRNA* network has proved that *miR-140-5p* could directly interact with GLUT1 *mRNA* and downregulate its expression, and in this manner, inhibit breast cancer cell access to glucose, thus preventing cancer cell growth (Figure 1) [38]. Another study stated that *miR-122* is a frequently produced miRNA and related to metastasis in breast cancer patients, which can directly interact with GLUT1. It has emerged that by virtue of its suppressing, GLUT1-induction mediated glucose uptake occurs, thereby inhibiting breast cancer metastasis [34]. In other cases, ncRNAs are not able to match directly with their targets, so they usually require an intermediary, such as transcription factors, to exert an effect on the key enzymes of metabolism. Interestingly, transcription factors usually interact with histone-modulating enzymes to regulate gene transcription. Acetyltransferases regulate histone acetylation and are often related to active transcription. Sine oculis homeobox 1 (SIX1) is a transcription factor and is reported to be a GLUT1 regulator in breast cancer. It can promote GLUT1 gene expression largely via histone acetyltransferases HBO1 and AIB1. Moreover, high SIX1 levels boost the indicator activity of GLUT1. In this regard, Li et al. observed that *miR-548a-3p* can diminish SIX1 expression through directly targeting its 3′-UTR, which leads to a notable decrease in the expression of GLUT1 in breast cancer. As a result, the overexpression of *miR-548a-3p* indirectly reduces GLUT1 expression, thus hindering breast cancer cell proliferation [39]. It was reported that *LINC00346* acts as an oncogenic, was overexpressed in breast cancer cells and that it exerts its oncogenic ability via the upregulation of GLUT1, thereby promoting breast cancer development [37]. It appears that *miR-148b* is a direct target of *LINC00346* and that *LINC00346* acts as a molecule that sponges *miR-148b* in breast cancer. *LINC00346* knockdown inhibits glycolysis and proliferation and conducts breast cancer cells towards apoptosis by upregulating *miR-148a* and *miR-148b*. The further study demonstrated that *LINC00346* upregulates *miR-148a* and *miR-148b*, thus, resulting in significant GLUT1 expression and predisposing breast cancer cells toward cell death [37]. In other words, some ncRNAs function in two steps; first, inducing their effect by affecting and regulating transcription factors, and second, indirectly altering the key enzymes of the metabolic process, such as GLUT1 expression [40,41].

### 5.2. ncRNA Relevance with the Key Enzymes of Glycolysis for the Efficient Regulation of Glucose Metabolism

Cancer cells, because of their rapid proliferation, need to shift or reprogram their metabolism from oxidative phosphorylation towards aerobic glycolysis by converting pyruvate to lactic acid instead of ending Acetyl-CoA particularly in the absence of sufficient oxygen which can be the classical instance of a metabolic reprograming pathway in cancers [42,43,44]. In this regard, some regulatory enzymes, including GLUTs, Hexokinase II (HKII), 6-Phosphofructo-2-kinase (PFKFB3), pyruvate dehydrogenase (PDH), lactate dehydrogenase A (LDHA) and pyruvate kinase M2 (PKM2), should be upregulated to accelerate the rate of glycolysis in cancer cells [45]. Several previous studies have displayed that some *ncRNA*s not only exert their anti-tumor ability by directly interacting with those enzymes but can also regulate indirectly by controlling the different transcriptional factor expressions such as p53, HIF-1 and c-MYC [39,46,47].

Hexokinases (HKs) catalyze the first and irreversible step of glycolysis [46]. It has four important isoforms, in which, HK2 is the prominent isoform in the cancer cells [47]. The malfunction of HK2 in breast cancer cells tends to be modulated using *ncRNAs* such as *miRNA* [48,49]. For instance, Kim et al. understood that *miR-155* is overexpressed by breast cancer cells for upregulating c-MYC and activating the phosphoinositide-3-kinase regulatory subunit alpha (PIK3R1)-PDK1/AKT-FOXO3a pathway to boost HK2 expression, thereby accelerating glycolysis and promoting breast cancer progression. Their further study cleared that losing *miR-155* represses cMYC, PIK3R1 and FOXO3a gene expressions which result in a considerable reduction in in vivo tumor growth via HK2 -mediated suppressing glycolysis [50]. Another report [48] indicated that *miR-155* can activate the signal transducers and activators of transcription 3 (STAT3), which has a binding site for HK2, hence comforting the transcription of HK2. On the other hand, it emerged that HK2 can directly bind to *miR-143* besides *miR-155* and modulate the glycolysis of breast cancer. Interestingly, *miR-143* is a negative regulator of HK2, yet the evidence proved that *miR-155* can inhibit *miR-143* by interacting with and CCAAT-enhancer binding protein beta (C/EBPβ), which results in a considerable upregulation of HK2 expression [48]. It was stated by Cao and colleagues that *CircRNF20* is a 499 bp transcript originating from exon 3-5 of the RNF20 gene. *circRNF20* is upregulated in breast cancer and has a key role in enhancing the glycolysis, glucose uptake, lactate production and ATP level which can lead to promoting breast cancer cell proliferation. Interestingly, it acts as a *miRNA* sponge which could target and reduce the level of *miR-487a*. Moreover, knocking down *circRNF20* can diminish the expression of *miR-487a* and hypoxia-inducible factor-1α (HIF-1α). Based on a bioinformatic prediction, *miR-487a* targets or binds the 3′-UTR of HIF-1α *mRNA*. Additionally, it is clear that HIF-1α can bind with location 1 of the HK2 promoter where it is an upstream promoter site of HK2 and facilitates the transcription of HK2, which accelerates glycolysis and promotes breast cancer. Overall, the knockdown of *CircRNF20* seems to be an efficient way to reduce *miR-487a*/HIF-1α/HK2-mediated glycolysis and the proliferation of breast cancer cells [51]. *CircRAD18A* originates from the *RAD18* gene and is highly expressed in triple-negative breast cancer. It facilitates the development of this disease by adjusting insulin-like growth factor 1 (IGF1) and fibroblast growth factor 2 (FGF2) expression [52]. More evidence exhibited that silencing *circRAD18A* as a *miRNA* sponge inhibits breast cancer cell viability, migration and invasion by hindering glycolysis and eventually promoting cell apoptosis. Mechanically, *circRAD18* can directly interact with *miR-613* and induce *miR-613*-associated suppressive effects on breast cancer cell progressive behaviors. Moreover, HK2 is a direct target of *miR-613*, and *circRAD18* positively modulates HK2 expression through sponging *miR-613*. In addition, *circRAD18* knockdown can repress tumor growth by upregulating *miR-613*. Furthermore, *miR-613* can match and bind within the 3ʹ-UTR of HK. Therefore, controlling *circRAD18* expression could be a helpful strategy in regulating the key enzymes of glycolysis and finally inhibiting the growth of breast cancer [53].

The second regulatory enzyme that relates to glycolysis is 6-Phosphofructo-2-kinase (PFKFB3) [54]. Its activity is required for cancer progression and migration [55]. The regulation of this enzyme can be performed by ncRNAs agents in various cancers such as reprogramed breast cancer. PFKFB3 silencing, via *ncRNAs*, not only limits the consequent outcome of the glycolysis pathway but also suppresses the cell proliferation and cell growth of reprogramed BC cells [56,57]. Upregulation of the *lncRNA LINC00538 (YIYA)* promotes glycolysis, cell proliferation and tumor growth in BC. *YIYA* is associated with the cytosolic cyclin-dependent kinase CDK6 and regulates CDK6-dependent phosphorylation of the fructose bisphosphates PFK2 (PFKFB3) in a cell cycle-independent manner. In breast cancer cells, these events promote the catalysis of glucose 6-phosphate (G6P) to fructose-2,6-bisphosphate/fructose-1,6-bisphosphate. *YIYA* KO leads to the accumulation of G6P and the depletion of F-2,6-BP/F-1,6-BP in cells, as well as decreasing cell proliferation, invasion and tumor growth. High *YIYA* expression levels accelerate CDK6-dependent PFKFB3 phosphorylation and lead to the enhanced phosphorylation of F6P to FBP [58]. It is suggested that PFKFB3 (6-Phosphofructo-2-kinase) is a direct target of *miR-206* in BC. Expressions of *miR-206* in BC decreased the PFKFB3 expression. PFKFB3 can produce fructose-2,6-bisphosphate (F2,6BP), as a critical activator of 6-phosphofructo-1-kinase (PFK-1) which accelerates the glycolysis process in BC. Therefore, suppressing PFKFB3 by *miR-206* reduces F2,6BP production and ATP generation that inhibits glycolysis, cell proliferation and migration in reprogramed BC cells [56]. *Circ_0102273* is mentioned as a regulator of PFKFB3 and is upregulated in breast cancer [59]. Yu et al. reported that silencing *circ_0102273* results in a significant inhibition of breast cancer cell proliferation and metastasis, largely by regulating and reducing glycolysis. Mechanistically, their further study reported that *miR-1236-3p* expression is promoted while PFKFB3 expression is diminished by silencing *circ_0102273*, which confirms *miR-1236-3p*-mediated interaction (indirect interaction) between *circ_0102273* and PFKFB3. Interestingly, *miR-1236-3p* interacts directly with *3′UTR* of PFKFB3 [59].

The final step of the glycolytic pathway is the conversion of phosphoenolpyruvate into pyruvate (Figure 1) for producing ATP, which is mediated by pyruvate kinase [60,61]. Pyruvate kinase 2 (PKM2) is the important and predominant isoform of pyruvate kinase in BC [62,63]. However, knockdown of it with ncRNAs could suppress NF–kB activity by demolishing the expression of the phosphorylated p65 protein [64]. In this regard, Xu et al. understood that PKM2 can directly interact with NF-κB p65 subunit to activate and regulate EGR1, miR-148a and miR-152 expression to overcome a signal that is crucial to maintain normal cell functions. In fact, they understood that EGR1 modulates both miR-148a and miR-152 expression through the binding of EGR1 to those miRNA gene promoters at different binding places, resulting in a complex regulation property by DNA methylation. This regulates tumor angiogenesis and cancer development and can be helpful in designing and discovering new treatments [28]. Signal transducers and activators of transcription 3 (Stat3) promote the upregulation of heterogeneous nuclear ribonucleoprotein (hnRNP)-A1 expression. *hnRNP-A1* can bind to pyruvate kinase isoenzyme (PKM) pre-messenger RNA. Downregulation of Stat3 leads to a higher glucose concentration and lower lactate concentration; these effects can be reversed by the expression of *hnRNP**-A1*. Stat3, as a transcription factor, is a functional target for *let**-7a**-5p* in BC cells and it is suppressed *by let**-7a**- 5p* to efficiently reduce the aerobic glycolysis [65]. In a study, the relevance between *lncRNA* ribonuclease P RNA component H1 (RPPH1) and *miR-122* was considered to clarify the regulatory role of PKM2 gene expression in BC cells, including MDA-MB-231 and MCF-7 [65]. It was shown that in BC cells overexpressing RPPH1, the cell cycle was promoted as well as cell proliferation and colony formation. It was demonstrated that there is an adverse correlation between the level of *miR-122* expression and RPPH1 overexpression. In addition, the same correlation is seen in the related knockdown model. It means that a downregulation of *lncRNA RPPH1* increases *miR-122* expression, which leads to an upregulation of the PKM2 gene expression (Figure 1). PKM2 expression and cell proliferation were reduced in BC cells transfected with mimic *miR-122* to overexpress RPPH [66]. Interestingly, it is understood that *miR-122* acts as a tumor suppressor through targeting *IGF1R* and modulating the PI3K/Akt/mTOR/p70S6K in breast cancer [67]. Accumulating evidence indicates that *circRNA-0000518* is highly expressed and upregulated in BC cells/tissue. In addition, it plays a key role in cancer progression via *miRNA-326*-mediated regulation of oncogenes or tumor suppressors [68,69]. For example, a study proved that *miR-1258* is predicated as a target of *circRNA-0000518* based on bioinformatics analysis [70]. Zinc finger E-box-binding homeobox 1 (ZEB1) is a transcription factor and is implicated in the promotion of the epithelial-mesenchymal transition (EMT) process and the development of different malignancies [71]. As predicted through bioinformatics analysis, ZEB1 was found to be a target for *miR-1258*. In addition, *miR-1258* tends to bind to the 3′-UTR of ZEB1. Based on this information, Liu et al. [70] studied the role of *circRNA-0000518* in BC progression. They realized that *circRNA-0000518* interference not only suppresses cell proliferation, migration and glycolytic metabolism but can also increase the apoptosis of BC cells. Additionally, they found that *circRNA-0000518* exerts its BC-progression effect by binding to *miR-1258* to induce ZEB1 expression. The expression of PKM2 was significantly diminished in *circRNA-0000518*-silenced BC cells, which confirms the special regulatory relevance between *circRNA-0000518* and cellular glycolysis (Figure 1).

Pyruvate dehydrogenase complex (PDC) catalyzes the oxidative decarboxylation of pyruvate and links glycolysis to the tricarboxylic acid cycle [72]. The mitochondrial pyruvate dehydrogenase (PDH) complex has an important regulatory role in the utilization of glucose. PDH, as a multi-enzyme complex, consists of multiple copies of three catalytic subunits, including E1 (pyruvate decarboxylase), E2 (dihydrolipoamide acetyltransferase) and E3 (dihydrolipoamide dehydrogenase), along with the E3-binding protein. Mammalian PDC activity is controlled by pyruvate dehydrogenase kinases (PDKs) that can phosphorylate and inactivate the PDC [73,74,75]. It is suggested that PDH is a novel target of some anti-cancer agents [75]. For instance, Eastlack et al., in their research, observed that pyruvate dehydrogenase protein x (PDHX) is a direct target of *miR-27b* as *oncomiR* in BC. PDHX is a structural component of the PDC that is required for the activity of PDC. *miR-27a* could upregulate PDHX and limit the convention of pyruvate to the tricarboxylic acid cycle (TCA); this leads to the accumulation of pyruvate and encourages the synthesizing of lactic acid which facilitates cell growth and the progression of breast cancer [76].

Pyruvate dehydrogenase kinase isoform 4 (PDK4) is a member of the PDK family, located in the mitochondrial matrix of cells, and can inhibit the entry of pyruvate into the TCA cycle by inhibiting PDH activity, thereby switching energy derivation to cytoplasmic glycolysis. Based on the previous report, *miR-211* can directly bind to the PDK4 mRNA sequence placed at the 302–308 site in the 3′UTR area. Its overexpression inhibits PDK4 and HIF-1A expression and conducts the breast cancer cells towards the energetic phase and radiosensitivity, respectively. In addition, overexpressing *miR-211* results in meaningful apoptosis by impacting on Bad, Bax, p53 and Bcl2 expression and inhibits stemness through downregulating octamer-binding transcription factor 3/4(Oct3/4), nanog, SRY-box transcription factor 2(SOX2), snail and Vascular endothelial growth factor A (VEGF-A) expression [77]. In another study, Peng et al. [78] reported that PDK1 is required for BC reprograming via the activation of glycolysis under hypoxic conditions. In this regard, they uncovered that *lncRNA H19* acts as a *miRNA let-7* sponge and has a key role in the progression of breast cancer. Mechanically, *lncRNA H19*, by impeding *let-7* activity, helps to release HIF1α which leads to PDK1 expression [78]. Intriguingly, it was indicated that PDK1 is a direct target of HIF1α [79]. This demonstrates that PDK1 is modulated via the *H19/let-7/HIF-1α*-signaling cascade as a metabolic switch for regulating glycolysis. Therefore, suppressing the PDK1 expression through *lncRNA H19* knockdown may be an efficient and rational strategy for reprogramed BC regulation [78].

Lactate dehydrogenase A (LDHA) catalyzes the conversion of pyruvate to lactate in the cytoplasm. As described above, cancerous cells produce a high level of lactate due to the high activity of lactate dehydrogenase [44,78,79]. In a cancer micro-environment, excess lactate creates an acidic condition that accelerates cancer progression and metastases [80]. Therefore, lactate, as a key metabolite, has a direct role in promoting cellular growth [81]. The effect of non-coding RNA as a metabolic switch for controlling cancer cell metabolisms have received notable attention, because they sometimes act as tumor suppressors and suppress the expression of regulatory enzymes [82,83]. For instance, it is indicated that *lncRNA-SNHG7* can regulate cell proliferation and breast cancer progression by inhibiting glycolysis. Further study details reported that c-Myc, as an oncoprotein or transcription factor, directly binds to the promoter area of *lncRNA-SNHG7* and positively regulates *lncRNA-SNHG7* expression. Mechanically, knocking down *lncRNA-SNHG7* results in a sharp diminishing in LDHA expression through interacting with and regulating *miR-34a-5p*. Therefore, downregulating c-Myc and *SNHG7* expression tends to be not only an efficient approach for regulating glycolysis but also a promising strategy to find new treatments for reprogramed breast cancer [84]. In another report, Xiao et al. studied the effect of *miR-34a* on the expression of LDHA. They observed that *miR 34-a* directly binds to the 3′-UTR of LDHA and downregulates its expression in breast cancer. Surprisingly, based on their conclusion, *miR-34a* inhibits glycolysis and induces cell proliferation in reprogramed BC by downregulating LDHA expression [85]. Fibroblast growth factor receptors (FGFR1) phosphorylate the LDHA to increase LDHA activity that improves the glycolysis process in BC [86] (Figure 1). In a study, based on its regulatory roles, *miR-361-5p* (known as a tumor suppressor) was employed to directly inhibit the FGFR1 expression, which acts as a promoter of key enzymes of metabolic. Ma et al. understood that FGFR1 returned the anti-glycolytic activity of *miR-361-5p* via upregulating the function of LDHA, which supports the mechanism that *miR-361-5p* hindered breast cancer cell glycolysis and cell proliferation. Furthermore, *miR-361-5p* can directly interact and bind 3′-UTR of matrix metalloproteinase-1 (MMP-1), inhibiting the invasion and metastasis of breast cancer cells. Moreover, *miR-361-5p* exerts its impact by binding to the 3′UTR FGFR1, declining the LDHA expression and regulating the balance between glycolytic metabolism and mitochondrial oxidative phosphorylation [87]. It has been demonstrated that the Janus-activated kinase (JAK) and the signal transducer and activator of transcription (STAT) pathways have critical roles in the various aspects of cancer prorogation, inflammation and immune response [88]. In a study, Lei and colleagues discovered that miR-155-5p interacts with a negative modulator of JAK/STAT signaling, similar to a suppressor of cytokine signaling 1 (SOCS1). Therefore, in the suppression state of miR-155-5p, SOCS1 can inactivate JAK/STAT signaling pathways and facilitate induction cell apoptosis [89]. Yin Yang 1 (YY1) is a zinc finger protein which exerts its regulatory effects by activating or inactivating gene expression according to chromatin structure and promoter background. YY1 acts as a transcription factor and is upregulated in different tumors, including those of BC. Interestingly, four types of *circRNA*, including *has–circ-0033169, has–circ-0033170, hsa–circ-0033171, hsa–circ-0033172* and *hsa–circ–0101187*, were discovered to originate from the YY1 gene. In a study, Zhang et al. [90] investigated a probable regulatory relevance between *circYY1* and *miR-769-3p* on the glycolysis process of BC. Their result appeared that *circYY1* promotes glycolysis and tumor growth through elevating *YY1* expression by sponging *miR-769-3p* in reprogramed BC. Further study indicated that the protein level of LDHA2 was notably downregulated in *circYY1*-silenced BC cells while it was increased in mimic *circYY1*-elevated BC cells. Furthermore, the cell viability, colony formation, migration and invasion, as well as the glycolysis, were constrained in *circYY1*-silienced and *miR-769-3p*-elevated BC cells, which were probably mediated by reducing the LDHA2 protein level [90] (Figure 1). It was proven that *miR-769-3p* blocks the Wnt/β-catenin pathway by targeting zinc finger E-box-binding homeobox 2 (ZEB2), thereby suppressing tumor growth [91].

The rapid transport of mono-carboxylates, such as lactate, across the plasma membrane of cancer cells is necessary for cell energy homeostasis [92]. However, this transporting is facilitated by proton-linked mono-carboxylate transporters (MCT) [93]. Mono-carboxylate transporter 1 (MCT1), also known as al-lactate transporter isoforms MCT1 and MCT4, are overexpressed in solid tumors [94] and play a very important role in cancer cell survival and proliferation with targeting potential in cancer treatment [95]. Inhibiting MCT1 activity via *non-coding RNA* or therapeutic agents may be an efficient manner to disrupt the homeostasis of cancer cells [96]. For this purpose, Sandra et al. reported that MCT1 is a direct target of *miR-342-3p*. Their result was that the overexpression of *miR-342-3p* negatively regulates the expression of MCT1 and disrupts the lactate fluxes. On the other hand, they stated that *miR-342-3p* is able to suppress the cell growth, proliferation and progression of BC cells through the direct inhibition of MCT1 expression [97]. Interestingly, it was reported that *miR-342-3p* can regulate glycolysis by modulating the key metabolic enzymes by repressing the IGF-1R-associated PI3K/AKT/GLUT1 signaling pathway, which indicates its importance as a cancer metabolic regulator [98]. In another study, Hou et al. mentioned that *miR-124* acts as a tumor suppressor and is upregulated in BC. It was proved that *miR-124* directly binds to the 3′-UTR of MCT1 which functions as a transporter of lactate. These data verified that *miR-124* could directly target Therefore, MCT1 is a direct target of *miR-124* in BC cells. The overexpression of *miR-124* suppresses the transporting of lactate and decreases the ATP production that leads to inhibit the cell proliferation, suggesting that *miR-124* may be served as an efficient therapeutic target against reprogramed breast cancer [99].

Generally, based on the reports discussed above, reprograming allows BC cells to achieve the essential elements that are needed for their rapid proliferation and progression. This could be performed by changing the molecular expression levels of enzymes that are involved in regulating metabolic pathways, such as the glycolysis process. However, by addressing the statement of explained studies, it can be concluded that following the adjustment of the reprogramed BC cells with *ncRNA* segments, the metabolic pathways of cells could return to a near-normal setting. It means that *ncRNAs* can rewire BC cell metabolism networks. Table 1 shows several segments of *ncRNAs* that exert their regulatory effects by controlling the glycolysis-involved enzymes in reprogramed BC cells.

## 6. *ncRNAs* Act as an Effective Regulator of Lipid Metabolism in Breast Cancer

Not only does glucose metabolism have an effective role in BC progression, but fatty acid metabolism, including fatty acid synthesis and fatty acid oxidation, also has an essential effect on the proliferation of BC cells [113,114]. Cells generate energy by breaking down fatty acid synthesis via fatty acid oxidation, also known as β-oxidation [115]. Fatty acid synthases are able to convert carbohydrates (glucose) and amino acids, such as glutamine, into metabolic intermediates, which are essential for cellular processes, including the maintenance and stability of the cell membrane structure with energy storing capability [116]. In cancer cells, the lipid metabolism is altered and that may affect numerous cellular processes, such as cell growth and proliferation [115]. Interestingly, some enzymes associated with fatty acid synthesis, including ATP citrate lyase, Acetyl-CoA carboxylase (ACC) and fatty acid synthase (FASN), are upregulated in some cancers. This allows cancer cells to achieve a wide range of precursor macromolecules that are essential for rapidly assembling new cancer cells. *ncRNAs* can be involved in the modulation of lipid metabolism in BC by targeting the expression or activity of regulatory enzymes.

ATP citrate lyase (ACL) is an important enzyme that cleaves citrate to generate oxaloacetate and Acetyl-CoA in various cancers such as BC. It acts as a key regulator between the higher rates of aerobic glycolysis and de novo lipid synthesis that appeared in several types of tumor cells [117]. In a study, Liu and colleagues revealed that there was an interplay between *miR-22* and ACL in BC tissue/cell lines (MCF-7). Moreover, it was discovered that ACLY plays a key role in signal transduction [118] and activates the Akt and PI3K/Akt signaling pathway, which is a key regulator in the modulating of tumor progression [119]. Liu et al. found that overexpressing miR-22 reduces ACLY expression by binding to the 3′-UTR regions of ACLY, which can block the PI3K/Akt signaling pathway and inhibit the growth and metastasis of breast cancer cells (Figure 2) [120].

Acetyl-CoA carboxylase (ACC) has a key regulatory effect on fatty acid metabolism and mediates malonyl-CoA formation through the carboxylation of Acetyl-CoA. In this regard, Singh et al. [121] showed the relevance of *miR-195* to cell proliferation, invasion and the metastasis process in BC cells, as well as the regulatory role of that miRNA on de novo lipogenesis-related key enzymes, such as ACC. It was demonstrated that *miR-195* could attenuate epithelial-mesenchymal transition (EMT) in BC cells through regulating the ACC expression level. Bioinformatics analysis displayed that there is a perfect complement between the 3′ UTR of ACC and the seed region of *miR-195*. On the other hand, the luciferase activity of ACC was increased in MCF-7 and MDA-MB-231 cells that were transfected with *anti-miR-195*, while such activity was reduced when cells were transfected with a *pre-miR-195* clone. The overexpression of *pre-miR-195* in BC cells led to a reduction in protein levels of ACC as well as co-transfection of *anti-miR-195* and *pre-miR-195* had upregulation in ACC protein expression, indicating that *miR-195* could directly target the key enzyme that regulates the de novo fatty acid synthesis pathway (Figure 2).

Fatty acid synthase (FASN), the single cytosolic enzyme applies for de novo synthesis of long-chain saturated fatty acids and is essential for cancer cell survival. Research shows that FASN is frequently expressed in different types of cancers, including BC [122]. Regulation of FASN expression by ncRNA may be an effective approach to control the reprogramed BC cell. *Circ-ADP* ribosylation factors such as *GTPase 8B* (*CircARL8B*) are located at chromosome 3 and originate from the exon of the gene ARL8B. It has been reported that *circARL8B* is unusually overexpressed in BC and its silencing can suppress cell viability, migration, invasion and fatty acid metabolism in that cancer [123]. *CircRNAs* can sponge *miR-653-5p* and regulates its function. High-mobility group AT-hook 2 (HMGA2) functions as an oncogenic protein and has been implicated in the development of various cancers, including breast cancer [124]. It can be used as a target for different *miRNAs*. In this regard, Wu et al. [123] aimed to clarify how *circARL8B* interacts with *miR-653-5p* or *HMGA2* to regulate BC through impressing the lipid metabolism-related key enzymes. They discovered that there was a negative correlation between *circARL8B* and *miR-653-5p* expression; indeed, *miR-653-5p* was the target of *circARL8B.* Furthermore, HMGA2 was found to be a target gene of *miR-653-5p* and through its 3′ UTR can interact with miR-653-5p. However, HMGA2 overexpression tends to abrogate the suppressive effect of *miR-653-5p* on BC cell progression. Interestingly, FASN expression was suppressed in BC cells transfected with *si-circARL8B* and in cells treated with *miR-653-5p*, while this suppressive effect was abolished when such cells were exposed to pcDNA-HMGA2, indicating that *circARL8B* exerts its FASN regulatory role by controlling *miR-653-5p* and HMGA2 expression levels (Figure 2). Emerging evidence shows that the possible mechanism of circARL8B can occur by activating the prostaglandin E2(PGE2)/PI3K/AKT/ glycogen synthase kinase-3 beta (GSK-3β)/Wnt/β-catenin-independent (β-catenin) pathway [123].

Carnitine palmitoyl transferase 1 (CPT1) is located in the external mitochondrial membrane and catalyzes the conversion of long-chain Acyl-CoA form to their relating long-chain acyl-carnitines for transportation into the mitochondria. CPT1 exists in three various isoforms and includes CPT1A, CPT1B and CPT1C [125]. The accumulating evidence suggests that all isoforms of these enzymes are not only expressed by different cancers cells [126,127] but also dysregulated in reprogramed cancer cells. Addressing previous reports, *non-coding RNA* has a regulatory effect on the CPT1A expressions enzyme. For instance, in a study [128], the regulatory effect of *miR-107* on the CPT1A expression enzyme was considered in reprogramed BC cells. The result proved that the suppressive effect of *miR-107* on CPT1 expression was mediated by *lncRNA* nuclear paraspeckle assembly transcript 1 (NEAT1). Indeed, *LncRNA NEAT1* increases the expression of CPT1A by suppressing *miR-107* by improving the progression and development of BC cells. Intriguingly, it is understood that overexpressing *LncRNA NEAT1* conducts breast cancer cells towards apoptosis by downregulating the cell cycle-related genes, such as cyclin D1 and cyclin-dependent kinase 4 (CDK4), and promoting the apoptotic-associated gene expression such as Bcl2-associated agonists of cell death (BAD), cysteinyl aspartate proteases (CASP9) and collagen type XVII a 1 (COL18A) [128]. In another study, Zeng et al. [129] discovered the relation between *miR-328-3p* and the CPT1A enzyme, which may be involved in the progression of BC. They found that *miR-328-3p* plays an important role in breast cancer growth and its expression has a negative correlation with BC cancer metastasis. Based on the output obtained from bioinformatics analysis, CPT1A indicated the highest interaction score with *miR-328-3p*, and it was found to be a downstream target for that *miRNA* in BC. It was observed that *miR-328-3p* could interrupt fatty acid metabolism and regulate fatty acid β-oxidation by controlling the CPT1A expression in reprogramed BC cells [129]. Overall, it emerged that miR-328-3p/CPT1A/fatty acid β-oxidation is responsible for breast cancer development and metastasis [129].

Cholesterol is the main precursor for steroid hormones, bile acid and other biological molecules. Abnormal changes in the metabolism of cholesterol can disrupt cell proliferation, growth, invasion and the migration of BC cells. This can display the pivotal role of cholesterol in the cell signaling pathways [130]. HMGCoA reductase (HMGCR) and HMG-CoA synthase 1 (HMGCS1), two main enzymes, mediate the modulation cholesterol biosynthesis/mevalonic acid (MVA) pathway and create a metabolic vulnerability, providing an opportunity to target to inhibit breast cancer. Bhardwaj et al. stated that *miR-140-3p* directly binds to the 3′UTR site of HMGCoA reductase (HMGCR) and HMG-CoA synthase 1 (HMGCS1) and suppresses the activity of those enzymes (Figure 2). Furthermore, they concluded that by activating AMPK, *miR-140-3p* can reduce HMGCR and HMGCS1 expression and prevent cholesterol biosynthesis through inhibiting the MVA pathway in triple-negative breast cancer (TNBC [131].

SQLE (squalene epoxidase) is a key enzyme used for cholesterol biosynthesis. In this regard, Qin and colleagues reported that *Lnc030* can increase SQLE *mRNA* stability and subsequently lead to an enhancement of cholesterol synthesis in BC. The study also revealed that there is a direct correlation between *Lnc030* and SQLE expression. In the other words, enhancing SQLE expression increases *Lnc030* levels which can boost the expression of some pluripotent transcription factors, such as c-Myc, Kruppel-like factor 4 (KLF4) and SOX2, which are helpful in maintaining breast cancer stem cells (BCSCs) stemness. Additionally, it was observed that Lnc030 cooperates with the poly(rC) binding protein 2 (PCBP2) to increase SQLE *mRNA* stability, which could explain how *Lnc030* elevates SQLE expression levels (Figure 2). *Lnc030*-involved mechanisms can eventually govern cholesterol synthesis, which has been mentioned, to modulate many oncogenic signals such as PI3K/Akt and Wnt/*β*-catenin. Subsequently, they concluded that the enhanced cholesterol synthesis stimulates the PI3K/Akt signaling activity, which is the basis of the maintenance progression of a breast tumor [132].

Generally, these findings proposed that the regulatory role of ncRNAs in BC cannot be denied. However, the investigations on the modulation of breast tumors affected by ncRNAs are still in their early stage, especially the effects of *lncRNA* and *circRNA* on key lipid-associated metabolic enzymes and their deep actions and reactions with transcription factors in breast cancer. Therefore, it can be concluded that targeting lipid metabolism-involved enzymes with *ncRNA* segments, including *lncRNA, miRNAs* and *circRNA*s, might be the most effective approach in regulating lipid metabolism malfunction. Table 2 lists different *non-coding RNA* relations with lipid metabolism-associated key enzymes.

## 7. *ncRNAs* Modulate Glutamine Reprograming in BC

Amino acids are essential components in all living cells. It is stated that some metabolic pathway amino acids, including glutamine, serine and glycine, are altered in reprogramed cancer cells such as those of BC [136,137]. Glutamine is one of the most abundant amino acids in the serum of the human body [138]. In normal cells, it is not a necessary amino acid, but in cancerous cells, glutamine is an essential substrate for the energy source for the generation of nucleotides, lipids and proteins [139,140]. All of the amino acids, including glutamine, need selective transporters to transport across the cell membrane [141]. Approximately 14 amino acid transporters exist to transport glutamine [136]. These transporters belong to four different solute carrier (SLC) families, including SLC1, SLC6, SLC7 and SLC38 [136]. The significance of glutamine in cancer originates from its capability in nitrogen and carbon, providing for a series of reactions that support cancer cell proliferation, invasiveness and metastasis. First, glutamine provides carbon sources for entry into the TCA cycle by generating α-ketoglutarate (α-KG). Second, glutamine also provides nitrogen for nucleotide and non-essential amino acid biosynthesis [136]. Glutamine-related enzymes are upregulated in TNBC cells to enhance glutamine uptake [136]. Indeed, glutamine provides a survival advantage for TNBC tumor cells; therefore, those tumor cells often exhibit a glutamine-dependent phenotype. Additionally, glutamine is converted to glutamate through the glutaminolysis process by mitochondrial glutaminases (GLS) [142,143]. According to bioinformatics databases, *miRNAs*, such as *miR-513c* and *miR-3163*, are involved in glutamine metabolism by downregulation of glutaminases as an important enzyme that catalyzes the conversion of glutamine to glutamate (Figure 2). Therefore, *miR-513c* as tumor suppressor may be most useful method for suppressing BC cells progression when it manically overexpressed [144].

Glutamate is able to trigger many different downstream pathways through the N-methyl-D-aspartate receptor (NMDAR), including guanylate-kinase-associated protein (GKAP) signaling which promotes the invasion of cancer cells [145]. In a study, Yin and colleagues [146] found that 17β-estradiol (E2) and GPER-specific agonist G1 can activate the G protein-coupled estrogen receptor (GPER), which diminishes *LncRNA-Glu* expression. On the other hand, *LncRNA-Glu* can specially bind vesicular glutamate transporter 2 (VGLUT2) and decrease VGLUT2′s transport activity in transporting glutamate. Interestingly, the GPER-associated decline of *lncRNA-Glu* boosts the transcriptional activity of VGLUT2, thereby facilitating the transportation of glutamate. Moreover, they observed that another way to enhance glutamate secretion is by activating the cyclic adenosine monophosphate(cAMP)–protein kinase A (PKA) signaling pathway via GPER. Indeed, *LncRNA-Glu*/VGLUT2 signaling cooperates with the cAMP–PKA signaling pathway to promote glutamate secretion in TNBC. This can lead to the activation of NMDAR and its downstream pathways such as calcium/calmodulin-dependent protein kinase (CaMK) and mitogen-activated protein kinase (MEK or MAPK), thus improving cellular invasion and metastasis. [146].

SLC7A11 is a subunit of the amino acid transport system that mediates the entry of cysteine into cells to convert glutamate [147]. The cysteine/glutamate xCT antiporter has a dominant manner for intracellular cysteine accumulation and this is essential for glutathione synthesis. Cysteine/glutamate xCT antiporter is expressed in approximately one-third of TNBC. Liu et al. have indicated that SLC7A11 3′-UTR has four binding sites for *miR-26b*. Therefore, *miR-26b* can directly modulate SLC7A11 expression through binding just two strong conserved sites in the SLC7A11 3′-UTR. They understood that the downregulation of *miR-26b* can contribute to the elevated expression SLC7A11, which mediates breast cancer growth and drug resistance. Consequently, it was concluded that *miR-26b* can suppress SLC7A11 expression and promote apoptosis by mainly targeting several apoptosis-mediated proteins, such as cyclin D2 (CCND2), cyclin E2 (CCNE2) and astrocyte elevated gene-1(AEG1) [148].

## 8. *ncRNAs* as a Potential Tool for Prognosis Metabolic Pathway Dysregulation in Breast Cancer

With the rapid development of genomics analysis, an increasing number of studies are demonstrating the alteration of *ncRNAs* in the progression and development of BC. Biomarkers have prognostic value and are used in predicting patient outcomes, disease advancement and relapse, during or after treatment [149,150]. Therefore, in most studies, researchers used diagnostic biomarkers to diagnose people with BC [149]. Today, the expression of ncRNAs in BC correlates with overall survival, metastasis and tumor grade; therefore, it can potentially be used as a biomarker for prognosis as they are involved in the signaling pathway and proliferation of malignant cells [151]. Moreover, ncRNAs can target various genes and modify their expression, which makes them useful candidates in the clinics for the prognosis and detection of BC [152,153]. For example, Xiong et al. concluded that the *long non**-**coding RNA nuclear paraspeckle assembly transcript 1 (NEAT1)* was frequently overexpressed in TNBC cells in comparison to the non-TNBC tissues/cells, which suggests that NEAT1 may be used as a good prognostic predictor to follow the therapeutic outcomes in TNBC patients [128]. MIR210HG displayed many effects during proliferation and abolished TNBC cells tumorigenic potential in BC. Based on the conclusion drawn by Du et al., MIR210HG expression predicted a poor prognosis for both relapse-free survival and overall survival in BC patients [107].

In a study, *miR-513c* and *miR-3163* were relatively downregulated in BC tissue samples in comparison to margin tissues. Therefore, *miR-513c* and *miR-3163* may serve as diagnostic and prognostic markers in patients with BC [144]. Chen et al. observed that the overexpression of *miR-22* prevents BC cell proliferation, colony formation, invasion and exerted cell apoptosis via the targeting of GLUT1. Additionally, they found that lower *miR-22* and higher GLUT1 expression levels are correlated with short disease-free survival times and poor overall survival in patients with BC, which indicates that *miR-22* and GLUT1 expression could be a valuable indicator of BC progression and prognosis [154].

Accumulating evidence suggests that *circRNAs* exert critical actions in tumor progression by sponging miRNA. In this regard, Liu and colleagues planned a systematic analysis to determine a *circRNA* that may be considered a good prognosis in BC. Based on their analysis identification, *CircYY1* was selected as sponger of *miR-769-3p* and had high expression in BC cells and tissues. However, it was concluded that a higher level of that *circRNA* was notably correlated with a poor prognosis of BC [90]. In a clinical investigation, *circRNF20* was highly expressed in the BC samples, and its overexpression is correlated with a poor prognosis in BC patients [51].

Addressing the obvious relation between *non-coding RNA* and the potential tools for the prognosis of BC, it can be concluded that *ncRNAs* are suggested to be employed not only as diagnostic biomarkers but also as a prognostic marker and predictive indicator for BC.

## 9. *Non-Coding RNA* as Therapeutic Targets for the Regulation of Reprogramed Metabolic Pathways of Breast Cancer

New insights into altered tumor metabolism have provided novel therapeutic strategies in the treatment of BC cells. The key role of *ncRNAs* in BC metabolism and its corresponded mechanisms increases the possibility of developing a *ncRNA*-base targeted treatment [155]. *miRNA, lncRNA* and *circRNA* mimics or inhibitors can be used to elevate or block the activity of metabolic-associated genes, which have a regulatory effect on cancer initiation or progression programs. Among them, *miRNAs* are often the extensively studied candidate, which act as either oncogenes or tumor suppressors and cause improper transnational prevention or destroy their target mRNA. miRNA-based treatments can be achieved through strategies such as regulation miRNA abnormal expression by replacing downregulated miRNAs or inhibiting upregulated miRNAs. Reestablishing the expression of miRNAs that are downregulated or specifically omitted in cells is the main goal of the replacement [156,157]. Moreover, restoring miRNA expression can occur via the infection of a viral carrier to stabilize a special miRNA or deliver mature miRNAs known as miRNA mimics, including double-stranded oligonucleotides of nearly 22 nt length, which bear a similar sequence to endogen mature miRNA [158]. In this regard, various types of non-viral delivery systems have been developed, which consist of a lipid-based vehicle, polymeric carrier, functionalized lipid or polymeric nanocarrier, nanovector with a positive, negative or neutral charged and inorganic materials [159,160]. In a study, Kardani and colleagues [161] aimed to inhibit *miR-155* expression in breast cancer cells through transfecting antagomir and AS1411 aptamer with gold nanoparticles (GNPs). They observed that the fabricated nanocarrier successfully transfers *antagomir-155* into the cells and this was evidenced by significant *miR-155* downregulation and considerable p53 *mRNA* expression as a direct target of *miR-155*. Indeed, suppressing *miR-155* inhibits cell proliferation and conducts breast cancer cells toward apoptosis by upregulating p53 mRNA [161]. As previously discussed, Kim et al. demonstrated that *miR-155* downregulation can prevent glycolysis metabolism through the downregulation of the HK2 enzyme and predispose breast cancer cells towards apoptosis (Figure 1) [50]. In another study, Alsadat et al. [162] developed a new cationic PEGylated niosomal formulation for *miRNA-34a* delivery to treat breast cancer. Their findings showed that *miRNA-34a*-loaded niosomes indicated enhanced cytotoxic activity against the breast cancer cells and tumor suppression, which demonstrated its promising antitumor impact on breast cancer cells [162]. On the same page, Xiao et al. showed that miR-34a directly interacts with LDHA to inhibit glycolysis. In fact, they proved that miR-34a overexpression downregulates the LDHA level, thereby impeding glycolysis and hindering cell proliferation and metastasis in reprogramed breast cancer cells [85]. A study, stated that the *miR-195* affects the inhibition of migration and invasion in BC cells via the disruption of lipid homeostasis. Therefore, *miR-195* could be considered as a novel therapeutic target for the effective treatment of BC (127).

Inhibiting lncRNA through RNA interference (RNAi) technology can be a helpful approach in lncRNA-based therapies against cancers like breast cancer. In this strategy, usually exogenous or mimic double-stranded RNAs, such as short interfering RNAs (siRNAs) and short hairpin RNAs (shRNAs), are employed for special gene knockdown [163,164]. This manner may also be applied to deliver exogenously produced lncRNA carriers into cancer cells to upregulate associated lncRNAs. Even though siRNA has more specificity to its target mRNA, its efficiency is ephemeral because of its instability, whereas shRNA can indicate a durable and long-range impact in vivo [165,166]. In a report, Zhang and colleagues used shRNA fragments to knock down *lncRNA RPPH1* expression in breast cancer cells. They observed obvious changes in breast tumor size in vivo after transfecting shRNA–*RPPH1*. In other words, silencing *lncRNA RPPH1* downregulates PKM2 gene expression by an indirect attachment with *miR-122* (Figure 1), which inhibits breast cancer cell proliferation and tumorigenesis and implies *lncRNA RPPH1′s* importance as a promising therapeutic target [66]. In another study, lentivirus-mediated *shRNA–SNHG7* transduction was performed for *lncRNA–SNHG7* knockdown in breast cancer cells. It emerged that suppressing *lncRNA–SNHG7* expression through *shRNA–SNHG7*-inhibited cell proliferation, by decreeing LDHA expression, resulted in a notable diminishment in glycolysis activity [84]. As discussed in earlier sections, the *lncRNA H19* [78], *lncRNA NEAT1* [128] and *lncRNA 030* [132] can exert their therapeutic roles through controlling PDK1, LDHA, CPT1A and SQLE, respectively. Therefore, knocking down or silencing those lncRNA using siRNA or shRNA can be a promising method for breast cancer treatments goals.

*CircRNAs* are involved in triggering and developing different tumors, including those of BC. It is reported that one of the primary activities of circRNAs is their function, as miRNA sponges and acts as an endogenous modulator of miRNA activity [167]. Xing et al. observed that *circRAD18* facilitates cell proliferation and glycolysis through the upregulation of HK2 by sponging *miR-613,* indicating that *circRAD18* could be valuable a therapeutic target for BC treatment [53]. In another study, it was confirmed that *circYY1* sponges *miR-769-3p*, reducing cell proliferation, migration and the invasion of reprogramed breast cancer cells through inhibiting LDHA2-reduction mediated glycolysis [90]. All of these studies show the important role of ncRNAs in different biological processes, triggering from the advent of cancer cell growth until they metastasize in a distant part. A sensitive insight about the special role of ncRNAs during diverse processes could help to attain a better understanding of malignancies such as cancer at the molecular level, facelifting the approach for the development of new diagnostic, prognostic and therapeutic innovations.

## 10. Conclusions

Metabolism reprograming is the hallmark characteristic of BC cells. In reprogramed BC cells, anaerobic glycolysis is mostly active to provide macromolecules that are essential for the rapid assembly of new cancer cells. In addition, lipid and glutamine metabolisms are changed to meet their needs. All these changes occur through a dysregulating expression of key enzymes that control metabolic pathways. Therefore, applying a strategy that uses the characteristics of tumor metabolism for reprogramed BC regulation and treatment might be promising. Although some metabolic modulating drugs have been used in clinical trials, the consequent results were not completely satisfactory. Researchers are now investigating to find certain therapeutic approaches that target the critical enzymes involved in different types of metabolic pathways. Therefore, it is crucial to study the relevance between *ncRNAs* and metabolic pathways, including glycolysis. There is a slight doubt that *ncRNA* has appeared as a versatile mediator in regulating the various aspects of BC metabolism. Several different members of *ncRNAs*, including *lncRNAs, miRNAs* and *circRNAs*, were found to play regulatory roles in BC metabolism, particularly in the glycolysis pathway, and in the metabolism of glucose, lipids and glutamine. Metabolic pathway regulation of BC by *ncRNAs* can enhance or suppress the tumorigenic process through the regulation of rate-limiting enzymes that mediate the glycolysis, lipids and glutamine metabolisms. Clarifying the special role of *ncRNAs* in BC with metabolism changes, may open a window for *ncRNA*-based regulators that are personalized and accuracy against BC. Interestingly, several *miRNAs* indicate promising outcomes in periclinal examination. However, the clinical application of *miRNA*-based therapies is limited by their unwanted and unessential interactions with targets that are not involved in the BC process. Unlike *miRNAs*, the relation of *lncRNAs* and *circRNAs* in BC, especially the mechanisms of breast tumor, to glucose, lipid and glutamine metabolism remains at an early stage and requires a plethora of further studies to establish their importance in treating BC. Since *lncRNAs* and *circRNA* can exert their regulatory effect to modulate the *miRNAs* involved in metabolic enzyme expressions, research to attain *lncRNAs* or *circRNA*-based therapies are very convincing. Overall, more studies into the mechanisms and actions of *ncRNAs* in BC could not only permit the finding of additional prognostic indicators, but could also be useful for the exploration of novel therapeutic methods.

## Figures and Tables

**Figure 1 cells-11-02973-f001:**
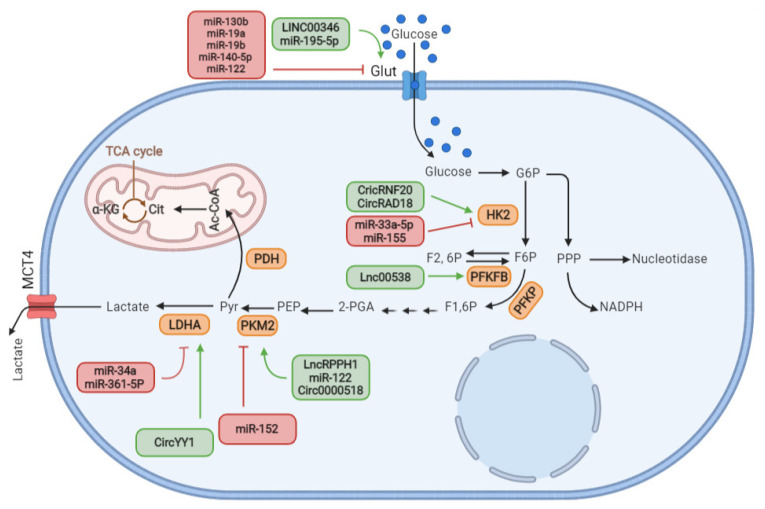
**Simplified schematic illustration of non-coding RNAs (ncRNAs) regulatory roles in glucose uptake and glycolysis.** In reprogramed cancer cells, glucose uptake is controlled by GLUTs located in the cell membrane and then converted to pyruvate under the influence of key enzymes in the glycolysis process. Interestingly, reprogramed BC cells modify glycolysis by altering the expression of key enzymes and undergo the conversion of pyruvate to lactate even when the oxygen level is adequate. Different types of ncRNAs are available to use to regulate the reprogramed BC metabolism. ncRNAs have exerted their cell metabolism regulatory effects through the modulation of the expression of GLUTs and key glycolysis enzymes, such as HK2, PFKFB, PKM2 and LDHA. The steps or enzymes that are upregulated or downregulated by ncRNAs are shown by the green and red arrows, respectively. Additionally, the related ncRNAs are listed in the semicircular shapes.

**Figure 2 cells-11-02973-f002:**
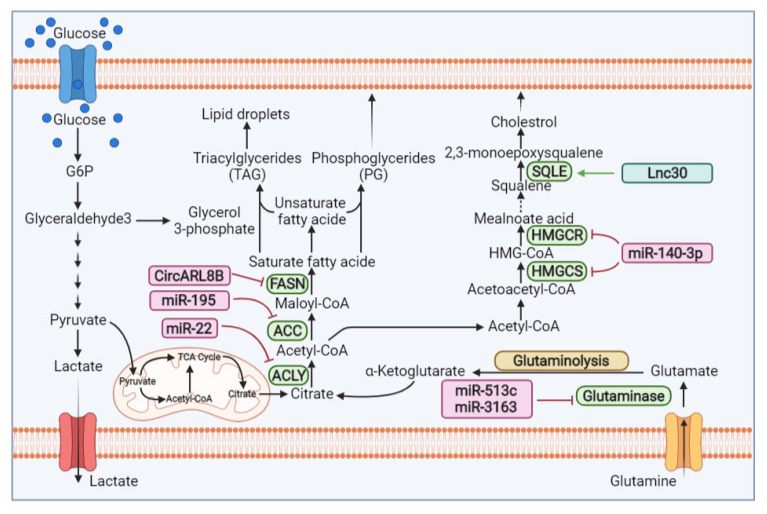
**Representation of the steps in the lipogenesis cycle, cholesterol and glutamine regulated by ncRNAs in reprogramed BC.** ncRNAs can modulate the molecules of lipids and cholesterol by regulating of the expression ACLY, ACC, HMGCS, HMGCR and SQLE enzymes. Moreover, ncRNAs induced their regulatory role on the conversation of glutamine to glutamate pathway via downregulation of glutaminases enzyme. The enzymes that are downregulated by ncRNAs are shown by the red arrow, whereas the upregulation effect of ncRNAs is illustrated by the green arrow. ACLY, ATP citrate lyase; ACC, Acetyl-CoA carboxylase; FASN, Fatty Acid Synthase; HMGCS, hydroxymethylglutaryl-CoA synthase; HMGCR, 3-Hydroxy-3-Methylglutaryl-CoA Reductase; SQLE, Squalene Epoxidase.

**Table 1 cells-11-02973-t001:** The interplay between ncRNAs and key enzymes of aerobic glycolysis in breast cancer.

Non-Coding RNAs	Target Enzyme	Function	Ref.
*miR-206*	PFKFB3	Downregulated	[56]
*miR-Let-7a-5p*	PFKM2	Downregulated	[65]
*MiR216b*	HK2	Downregulated	[100]
*miR-23a/24/210*	LDHA, LDHB	Downregulated	[101]
*miR-30a-5p*	LDHA	Downregulated	[102]
*IncRNA MAFG-AS1*	HK2	Upregulated	[103]
*lncRNA UCA1*	HK2	Upregulated	[104]
*lncRNA BCAR4*	PFKFB3	Upregulated	[105]
*lncRNA FGD5-AS1*	PKM2	Upregulated	[106]
*lnc RNA MIR210HG*	LDHA	Upregulated	[107]
*lncRNA HISLA*	LDHA	Upregulated	[108]
*lncRNA FGF13-AS1*	PDK1	Downregulated	[109]
*Circle RNA circABCB10*	LDHA	Upregulated	[108]
*CircRNA DENND4C*	HK2	Upregulated	[110]
*miR-Let-7a*	PKM2	Downregulated	[111]
*Hsa_circ_0069094*	HK2	Upregulated	[111]
*circHIPK3*	HK2	Upregulated	[112]

**Table 2 cells-11-02973-t002:** Summaries of *ncRNAs* that exert their regulatory effect by modulating lipid metabolism key enzymes in reprogramed BC.

Non-Coding RNA	Target Enzyme	Metabolic Pathway	Function	Ref
*miR-22*	ACL	Upregulated	Fatty acid metabolism	[133]
*miR15-1a, miR16-1a*	FASN	downregulated	Fatty acid metabolism	[134]
*lncRNA AGAP2-AS1*	CPT1	Upregulated	Fatty acid oxidation	[135]

## Data Availability

Not applicable.

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
