# Peer review of "Metabolic Pathways in Breast Cancer Reprograming: An Insight to Non-Coding RNAs"

_cells, 2022, doi:10.3390/cells11192973_

Round 1
Reviewer 1 Report
The authors summarize the recent notions on RNA activity in breast cancer. The different metabolic pathways are taken into consideration, with a particular interest in glycolysis, the pathway of lipids and glutamine.
The authors have completed the description of the metabolic pathways , typical of a biochemistry book, with some information on the regulation of the different key enzymes by different molecules of RNA
In my opinion, they should reduce the mere descriptive part of enzyme activity to broaden the description of how these RNA molecules impact and regulate breast cancer. Let me explain they should not limit themselves to describing that miRNA reduces the expression of an enzyme but how this can regulate the biology of breast cancer.
Also they have to discuss the potential use of RNA molecules in cancer therapy
Author Response
Reviewer #1:
The authors summarize the recent notions on RNA activity in breast cancer. The different metabolic pathways are taken into consideration, with a particular interest in glycolysis, the pathway of lipids and glutamine.
The authors have completed the description of the metabolic pathways, typical of a biochemistry book, with some information on the regulation of the different key enzymes by different molecules of RNA
Response: Thank you very much for the compliment and for pointing out your valuable comment.
In my opinion, they should reduce the mere descriptive part of enzyme activity to broaden the description of how these RNA molecules impact and regulate breast cancer. Let me explain they should not limit themselves to describing that miRNA reduces the expression of an enzyme but how this can regulate the biology of breast cancer.
Response; Thank you very much for your nice comment. The authors believe that the suggested comment can add scientific points to our manuscript. In this regard, the authors have reduced some parts of metabolic-related key enzymes explanation and added a notable deal of information for finding out how ncRNAs govern metabolic enzymes and how they interact with their upstream and downstream genes to modulate the biology of breast cancer. Yet, some of the discussed examples in an earlier version of the manuscript had shown helpful insight about ncRNAs and the regulation effect of them on key enzymes of metabolic by interacting with some effectors like transcription factors and impressing breast cancer malignant behavior such as cells growth, proliferation, invasion and metastasis. To see your suggested comment please see the highlighted lines 146-152, 154-160, 162-171, 172-179, 180-181, 197-200,210-223,228-233, 23-247,27-274,279-286, 302-304, 312, 337-344, 345-350, 361-368, 376-382, 383-389, 403-404, 417-419,425-426,459-465, 03-504, 509-511, 523-528, 536-537, 541-544,546-549, 553-557, 567-568, 604-617 and 622-629 in revised version of manuscript.
Also, they have to discuss the potential use of RNA molecules in cancer therapy
Response: Thanks for your suggestion. The authors made the suggested comment and added some new points for further illuminating and clarifying the possible effect of RNA molecules in cancer therapy. To observe the new changes please see the highlighted lines of 675-705, 708-720, 721-725, 727-729, 731-732 and 735-737 in a revised version of the manuscript.
Reviewer 2 Report
This is a very well written review article explaining the importance of metabolic dysfunction in cancer and its regulation by non-coding RNAs. There are some suggestions to further improve this article-
1. This review article is supposed to be focused on non-coding RNA-mediated metabolic dysregulation in breast cancer (BC) but BC specific examples of nc-RNA regulating metabolic genes which leads to metabolic reprogramming are lacking throughout the text.
2. Another major shortcoming is Authors cited other review articles a lot which is not appreciable for a review article. For example- in Section 4, authors cited 4 articles in support of their statements but 3 are of other review articles.
3. Authors discussed the roles of metabolic genes in detail. Rather authors should include the role of nc-RNAs in regulation of those genes and then discuss the effect on overall metabolism and cancer growth.
4. Page 2 Line 72- lncRNAs are more than 200 nts. Please correct the statement.
5. Page 2 Line 81- How circRNAs serves as transcription factors?
6. Page 3 Line 98- ‘dysregulation in metabolic pathways develop cancer’. This sentence is misleading and should be opposite else authors should give some examples from the literature to support this.
Author Response
This is a very well-written review article explaining the importance of metabolic dysfunction in cancer and its regulation by non-coding RNAs. There are some suggestions to further improve this article.
Response: Thank you very much for your compliment.
Comments:
- This review article is supposed to be focused on non-coding RNA-mediated metabolic dysregulation in breast cancer (BC) but BC-specific examples of ncRNA regulating metabolic genes which leads to metabolic reprogramming are lacking throughout the text.
Response: Thanks a lot for putting in your comment. The authors tried to picture the main effectors that resulted in metabolic reprogramming. Having said that, as you know, some enzymes have direct interaction with some ncRNAs and by virtue of altering the expression of metabolic enzymes or changing ncRNAs expression levels accelerating cancer cells toward reprogramming. On the other, in the revised version of the manuscript, the authors provided some explanations that can help to find out how the ncRNAs modulating genes govern the metabolic reprogramming in breast cancer cells. To observe the added information please see the lines of 162-167, 172-179, 210-223, 228-233, 235-247, 267-274, 279-286, 302-304, 312, 337-344, 345-350, 361-368, 376-382, 383-389, 403-404, 417-419, 459-465, 503-504, 509-511, 523-528, 536-537, 541-544, 546-549, 553-556, 559-562, 604-617 and 622-629 in the improved version of manuscript.
- Another major shortcoming is Authors cited other review articles a lot which is not appreciable for a review article. For example- in Section 4, the authors cited 4 articles in support of their statements but 3 are from other review articles.
Response: Thanks for your helpful comment. Based on your suggestion we used original articles instead of review reports. Please see the corrected references in lines 104, 109 and 115 in a revised version of the manuscript.
- Authors discussed the roles of metabolic genes in detail. Rather authors should include the role of ncRNAs in the regulation of those genes and then discuss the effect on overall metabolism and cancer growth.
Response: Thank you very much for your valuable suggestion that absolutely adds a great deal of scientific points to our manuscript. In the revised version of manuscript, the authors made your suggestion on most of discussed examples and you can see the new points by looking at lines of 162-167, 172-179, 210-223, 228-233, 235-247, 267-274, 279-286, 302-304, 312, 337-344, 345-350, 361-368, 376-382, 383-389, 403-404, 417-419, 459-465, 503-504, 509-511, 523-528, 536-537, 541-544, 546-549, 553-556, 559-562, 604-617 and 622-629 in the corrected form of manuscript.
- Page 2 Line 72- lncRNAs are more than 200 nts. Please correct the statement.
Response: Thanks for your valuable proposition. Your suggestion has been revised please see lines 70-72.
- Page 2 Line 81- How circRNAs serve as transcription factors?
Response: Thank you very much for your more detailed question. As you know, there is a low understanding of the transcriptional regulator role of circRNAs. Yet there are some reports that display circRNAs have a transcriptional regulatory role. It is hypothesized that one of the main functions of circRNAs is their action as miRNA sponges, they are acting as endogenous modulators of miRNA activity. As it has been proved that miRNAs are implicated in post-transcriptional regulation of gene expression by matching with their complementary mRNAs and stimulating translational inhibition through degradation or p-body storage of the mRNA. Therefore, circRNAs by virtue of sponging the miRNAs play a key role in regulating gene expression. In fact, they act as crucial post-transcriptional modulators and play important role in a regulatory network including circRNAs, miRNAs and mRNAs. Additionally, intron-containing circRNAs are remained in the nucleus and are supposed to regulate the transcription of genes. It is reported that the circRNAs can upregulate their parental gene expression through the formation of a circRNA-U1 small nuclear (sn) ribonucleoprotein complex that tends to interact with RNA polymerase II. Nuclear circRNAs can also interact with additional transcriptional and splicing regulators, involving U1A and U1C. Interestingly, it has been shown that these circRNAs localize to the place of their parental genes and co-immunoprecipitate with their promoters and regulate their expression. Furthermore, some circRNAs may affect the nuclear translocation of other proteins to the nucleus and the regulation of gene transcription. For example, CircAmotl1 may increase nuclear translocation of signal transducer and activator of transcription 3 (STAT3) for regulating the expression of mitosis-related genes. All in all, thanks to your more informative question the author decided to change it in the revised version of the manuscript. Please look at lines 81-82 to view the revision form of your question.
- Page 3 Line 98- ‘dysregulation in metabolic pathways develop cancer. This sentence is misleading and should be the opposite else authors should give some examples from the literature to support this.
Response: Thank you very much for pointing out the potential for misunderstanding. The authors are completely agreeing with the respected reviewer's comment. In accordance with your comment, we changed that sentence to reach the well-mean. Please see it in lines 97-99 in the highlighted part of the revised version of the manuscript.
Round 2
Reviewer 1 Report
The Authors revised the manuscript following the referee's suggestions. The manuscript can be published in the present form.